# A Robot Object Recognition Method Based on Scene Text Reading in Home Environments

**DOI:** 10.3390/s21051919

**Published:** 2021-03-09

**Authors:** Shuhua Liu, Huixin Xu, Qi Li, Fei Zhang, Kun Hou

**Affiliations:** School of Information Science and Technology, Northeast Normal University, Changchun 130117, China; Liush129@nenu.edu.cn (S.L.); xuhx623@nenu.edu.cn (H.X.); liq821@nenu.edu.cn (Q.L.); zhangf761@nenu.edu.cn (F.Z.)

**Keywords:** robot object recognition, complex scenes, scene text detection, text recognition

## Abstract

With the aim to solve issues of robot object recognition in complex scenes, this paper proposes an object recognition method based on scene text reading. The proposed method simulates human-like behavior and accurately identifies objects with texts through careful reading. First, deep learning models with high accuracy are adopted to detect and recognize text in multi-view. Second, datasets including 102,000 Chinese and English scene text images and their inverse are generated. The F-measure of text detection is improved by 0.4% and the recognition accuracy is improved by 1.26% because the model is trained by these two datasets. Finally, a robot object recognition method is proposed based on the scene text reading. The robot detects and recognizes texts in the image and then stores the recognition results in a text file. When the user gives the robot a fetching instruction, the robot searches for corresponding keywords from the text files and achieves the confidence of multiple objects in the scene image. Then, the object with the maximum confidence is selected as the target. The results show that the robot can accurately distinguish objects with arbitrary shape and category, and it can effectively solve the problem of object recognition in home environments.

## 1. Introduction

Accurately perceiving and recognizing the scene object in home environments is a challenge of robotics technology. Scene text reading can improve the robot’s object recognition ability, path planning, and real-time visual translation. The rapid development of robotics and the arrival of the aging society increase the significant demand for home service robots. Fetching is one of the general tasks, and furthermore, object recognition with high accuracy is essential because of foods and medicines. At present, the object recognition method based on deep learning mainly identifies object shapes. However, fetching household objects is challenging because of the demand of recognizing different shape objects in the same category, or similar shapes from different categories. Therefore, recognizing objects only by shape is insufficient.

This paper proposes an object recognition method based on scene text reading, which enables the robot to perceive the environment more efficiently. The robot detects and infers an object by reading its text. Rather than simply recognizing the object based on shape, the robot acts similar to a “cultured person” who can read the text. We will give a detailed description to the robot recognizing objects according to human instructions in Section 3. The main technology of this object recognition method is scene text reading; therefore, we improve the precision of text recognition firstly and then perform a series of experiments on robots.

Scene text reading includes text detection and recognition, which is becoming mainstream technology for deep learning because of its excellent performance. The generalization ability of the object detection model allows for achieving the task of text detection. The Region Convolution Neural Network (R-CNN)-based object detection model [1,2,3] and its improved version trade speed for accuracy. The results of high mean Average Precision (mAP) makes R-CNN a general and basic model of text detection. Mask R-CNN [4] extends R-CNN detection methods and is applied to image segmentation. By comparison, detection models such as You Only Look Once (YOLO [5,6,7]) and Single Shot MultiBox Detector (SSD [8]) improve detection speed at the expense of slight accuracy. The ingenious trade-off between speed and the accuracy leads to the popularity of these models. Text recognition is responsible for recognizing texts in detection regions, and it usually adopts CNN, Long Short-term Memory (LSTM), and attention models.

Scene images are usually blurred, distorted, covered with large shadow, or low resolution. At the same time, the enormous variety of fonts, directions, and sizes of scene text increases the difficulty of its detection and recognition. Multilingual recognition is also a large challenge for scene text recognition. Apart from digits and symbols, this paper deals with Chinese and English recognition. Compared with English, Chinese characters have more complex structures with often the same or similar strokes, which further increase the difficulty of recognition. The above factors lead to the unsatisfactory effect of most text recognition models. Figure 1 shows several examples, in which green boxes are detection marks and yellow fonts are the recognition results. Figure 1a–c represent a blurred image, different directions, and Chinese complicated and diverse fonts, respectively.

Based on the above reasons, computer vision and natural language processing technologies have been combined [9]. Thus, scene text reading is regarded as an object detection and language model based on sequence transduction. This approach effectively solved the problem of text shape. The framework is based on the Cascade R-CNN [10], which can detect scene text in any shape. During the recognition, context information is combined with Word2vec [11] based on the attention mechanism, which is helpful for reading scene text. A large number of experiments have proven that the framework effectively improves the reading ability of scene text.

The main contributions of this paper consist of three aspects, as follows:(1)This paper proposes an object recognition approach based on scene text reading. With this approach, robots can recognize arbitrary-shape and arbitrary-category objects. In addition, this approach speeds up robot mimicking of human recognition behavior.(2)To improve the recognition accuracy of models from reference [9], we generate a new dataset and its inverse. The generated dataset contains 102,000 images with labeled documents, while the inverse dataset inverts the pixel value of generated images without changing the labels. After training on these datasets, the recognition accuracy of the model is improved by 1.26%.(3)Experiments are carried out on relations between confidence thresholds of text boxes and recognition effect. A higher confidence threshold results in more accurate recognition. However, useful information may be missed. By statistics of test samples, the confidence threshold is set at 0.97, which is a good balance that indicates that the key information is reserved and the recognition accuracy is high with few wrong words.

## 2. Related Works

In recent years, mobile robot object recognition has become popular. Object recognition includes object detection and recognition. Robot object detection focuses on improving precision in particular scenarios or on finding solutions to obtain state-of-the-art accuracy. Maiettini et al. proposed a hybrid architecture that integrates Faster R-CNN with an optimized kernel-based method [12] for region classification [13]. It naturally trains a robot to detect novel objects in a few seconds. In reference [14], Self-supervised Sample Mining (SSM) was extended to enable online adaptive object detection for robotics by integrating the Weakly-supervised Learning (WSL) sample selection strategy with the online object detection method [15]. Ceola et al. improved the feature extractor of reference [13] by substituting Faster R-CNN’s first layer with Mask R-CNN’s first layer [16]. It extended the object detection system with the fast learning approach, improving speed and accuracy. Maiettini et al. presented an empirical analysis of different Active Learning (AL) techniques in a challenging robotic scenario, proposing a solution for the Self-supervised Learning (SSL) failure cases under domain shift and improving the overall labeling efficiency [17]. On robot object recognition, shape, color, and texture are often employed as main features. Browatzki et al. adopted Pyramids of Histograms of Oriented Gradients feature (PHOG) in robot object recognition to capture the shape information and color histograms to describe the basic appearance [18]. Alam et al. proposed a simultaneous recurrent network (SRN) auto encoder based on a deep learning algorithm for the Nao robot to recognize objects [19]. With the SRN, Nao can recognize human faces and images of 26 letters from A to Z. The number of training parameters is significantly reduced by sharing weights in the hidden layer. This model has better recognition performance than the general five layers stacked auto encoder (SAE). Yoshimoto et al. proposed a new architecture “Dual Stream-VGG16 (DS-VGG16)” to train RGB-D images taken by a Kinect camera [20]. By adding depth to the image, the recognition accuracy is improved up to 99.9%. Subsequently, based on the proposed method, they developed an object recognition system with an interface of robot operating system for integrating into service robots. Chen et al. also fed RGB-D images into a Faster R-CNN network to realize the recognition and localization of present objects out of 50 different classes [21]. Fu et al. proposed a fog computing with object recognition system based on YOLO v3 to allow robots to perceive and recognize in Internet of Things (IoT) [22]. The proposed scheme significantly achieves state-of-art performance compared with the YOLO v2, and allows the robot to recognize objects in real time.

Several human assistance-based robot object recognition methods have been developed. Takaki et al. proposed a system with a mobile robot that uses cooperative object recognition [23], moves to a spot where a user wishes to go, and transfers images containing various objects in the spot. The user chooses one of those objects and sends a request to the robot to get it. Several objects are found on a captured image by matching their template images with several sizes. Cartucho et al. used an approach that combines two neural nets YOLO v2 [6] and HUMAN to recognize all objects in the image. Humans were employed to answer several questions with “yes” or “no” to assist the object recognition [24]. Kasaei et al. [25] propose a system that adapts to unseen environments by incrementally learning new object categories from human interactions. Eriksen et al. [26] use human supervision to acquire only domain-relevant data, reducing training data requirement. Venkatesh et al. [27] also improves robots object recognition ability with the help of humans. They propose a spatial attention modulation mechanism and teach a robot novel objects it has not encountered before by pointing a hand at the new object of interest.

Different from all of the above literature, this paper proposes a robot object recognition method based on scene text reading. The robot recognizes objects by reading its texts and not by shape, color, or other information. Furthermore, our method does not need human assistance to recognition. Experiments show that the problem of confusion for similar-shape objects is solved, and the recognition accuracy is greatly improved.

## 3. Methodology

Figure 2 shows the framework for the robot to recognize objects according to human instructions. The robot transforms keywords of human instruction into texts by speech recognition and then identifies texts on every object. When the result of the current object includes keywords from the human instruction, the robot stops the recognition and takes the object, and it otherwise continues to recognize the next object.

To recognize texts on objects, the robot needs to carry out text detection and recognition. The recognition model [9] is adopted, as shown as in Figure 3. A scene image is fed into the text detection module. First, Cascade Mask R-CNN detects the text region and marks the bounding box in green. In the following, the mask branch of Mask R-CNN segments the text regions. Finally, the masked images are fed to the attention-based text recognition module for sequence classification. InceptionV4 [28] is used as the backbone of the recognition module, which aligns visual context with corresponding character embedding in a semi-supervised optimization method. Then, the detected region and recognized texts are marked together on the input image to obtain the final result. The following sections briefly describe text detection and recognition.

### 3.1. Text Detection

With Cascade Mask R-CNN as the main body of the detection framework, the instance segmentation task is completed by the mask branch of Mask R-CNN. A segmentation branch is performed at each cascade stage to maximize the diversity of samples used to learn the mask prediction task. Figure 4 shows the detection module, where “conv”, “pool”, “Head”, “B”, and “C” represent the input image, backbone convolutions, region-wise feature extraction, network head, bounding box, and classification, respectively. Moreover, “S” and “B0” denote a segmentation branch and proposals in all architectures.

The text detection module based on Cascade Mask R-CNN consists of two components, detection and segmentation. In the latter, the goal of optimization is achieved by minimizing the loss function $L_{total}$ expressed as:(1)$$\;\;\;\;\;smooth_{L_{1}}\left(x\right)=\left\{\begin{array}{c} 0.5x^{2}\;\;\;\;\;\;\;\;\;\;\;\;\;\;\;\;\;\;if\left|x\right|<1\\ \left|x\right|-0.5\;\;\;\;\;\;\;\;\;\;\;\;\;\;otherwise\; \end{array}\right.$$
(2)$$L_{Ca}=\sum_{i=1}^{N}\left(L_{Fast}\left({\hat{y}}_{i},y_{i}\right)+L_{1}\left({\hat{b}}_{i},b_{i}\right)\right)$$
(3)$$\;L_{total}=L_{RPN}+L_{Mask}+L_{Ca}+\lambda L_{2}$$
where $L_{1}$ represents the smoothed $L_{1}$ loss in Fast R-CNN [2]; in Equation (2), $L_{Ca}$ is the sum of multiple levels cross-entropy loss $L_{Fast}$ and $L_{1}$; N is the number of multiple cascade stages; ${\hat{y}}_{i}$ is the label logarithm and $y_{i}$ is the one-hot ground truth;$\;{\hat{b}}_{i}$ is the estimated bounding box transformation parameter and $b_{i}$ is the ground truth. In Equation (3), $L_{total}$ represents the total loss of the segmentation component and it is the goal of optimization; $L_{RPN}$ represents the loss function of the Region Proposal Network (RPN) network; $L_{Mask}$ from Mask R-CNN [4] represents the average binary cross-entropy loss; λ represents the weight decay factor; and $L_{2}$ represents the L_2_ regularization loss to prevent overfitting. 

### 3.2. Arbitrary-Shaped Scene Text Recognition

The use of RNN has achieved remarkable success in the horizontal and clear text recognition. By contrast, the challenge remains in recognizing irregular text in any direction, such as fan-shaped, oblique, and vertical. Most text recognition models cannot handle irregular texts of arbitrary shapes in the scene text. Figure 5 shows horizontal and irregular scene texts, such as oblique and distorted text.

The Bahdanau attention mechanism [29] introduced to the recognition model to improve the accuracy of predicting words, owing to its focus on the information region. In view of the relevance of scene text recognition and Natural Language Processing (NLP), the model combines the technique in computer vision and NLP to effectively solve the problem of irregular text shapes. Figure 6 shows the text recognition model. 

The text image is fed into CNN to extract features map *V*; then, the attention weight $a_{t}$ is calculated according to Equation (4). $I_{t}$ is the context vector by calculating feature vector *V* and attention weight $a_{t}$ by Equation (5). The character probability is calculated according to Equation (6) and the predicted word embedding ${\hat{e}}_{t}$. Repeat this process until all characters are recognized. The detailed explanation of this model is presented in Reference [9]. The LSTM is improved by adding the current attention to analyze the feature sequence.
(4)$$\;a_{t}=AttentionFunction\left(V,h_{t-1}\right)$$
where V is the feature map of the CNN network, and $h_{t-1}$ is the hidden state of LSTM in time step *t* − 1.
(5)$$I_{t}=\sum a_{t}.V,\;1\leq t\leq T$$
where $a_{t}$ is the attention weight in time step *t*, and *T* represents the maximum sequence length.
(6)$$P=\prod_{t=1}^{T}P(e_{t}|h_{t-1},I_{t},{\hat{e}}_{t-1})$$
where $e_{t}$ represents the embedding in the global word embedding matrix E corresponding to time step *t*. *T* represents the maximum sequence length. $h_{t-1}$ is the hidden state of the last LSTM unit,$\;I_{t}$ is the context vector, and ${\hat{e}}_{t-1}$ is the expectation of previous embedding. The initial hidden state $h_{0}$ of the LSTM is the same as [1,30].

## 4. Experiments on Detection and Recognition Models

### 4.1. Dataset

In the experiments, ICDAR2019 Art [31], LSVT [32], and ReCTs [33] generate the text dataset and its inverse that are used for training. Verification is performed on the training set of RCTW-17 [34], and the test set of RCTW-17 serves as the test set. RCTW-17 is a common dataset used in scene text detection and recognition tasks. The contents include 8034 annotated training and validation images and 4229 test images, such as street scenes, posters, menus, indoor scenes, and screenshots. The text shape is multi-directional, and the labeling method is quadrilateral.

To further improve the accuracy of the recognition model, we generate a text dataset with 102,000 annotated text images. This dataset includes 71,000 simplified and traditional Chinese text images, 21,000 uppercase English images, and 10,000 lowercase English images. The images are all black text on a white background. The label format is the same as RCTW-17, but the text lines are only horizontal. The images are divided into four categories: standard, oblique, distorted, and obscure. The random generation of Chinese texts include one to 10 words, while the English ones include one to five words. Table 1 shows the specific divisions. Figure 7 shows an example of the generating dataset.

### 4.2. Parameter Settings

The text detection module uses the Cascade Mask R-CNN network with ResNet101 as the backbone. Therefore, its parameters are set to the default values of Cascade R-CNN and Mask R-CNN. For data augmentation, several operations such as rotation, translation, and cropping are performed on the input image. The minimum crop size is set to 224. Stochastic Gradient Descent (SGD), with a learning rate of 0.01 and momentum of 0.9, is used as the optimizer.

For the text recognition module, the Adam optimizer is used with learning rate and weight decay set to 0.0001 and 0.00001, respectively. The model Pre-train InceptionV4 on ImageNet serves as initialization.

### 4.3. Text Detection Results

The common evaluation indicators for text detection are Recall, Precision, F1-measure, and Average Precision (AP). F1-measure and AP are comprehensive evaluation indicators. The detected text region coordinates and confidence are submitted to the RCTW-17 competition platform (https://rctw.vlrlab.net/08/03/2020, accessed on 8 March 2020). The F-measure, precision, and recall obtained on task1; that is, detection task of RCTW-17 are 0.6817, 0.7424, and 0.6302, respectively, as shown in Figure 8. The AP is 0.5664. Table 2 lists the results comparison between our algorithm and state-of-the-art methods on the dataset RCTW-17. The recall rate of the proposed method reaches 63%, which is higher than those of other methods. The accuracy rate of the proposed algorithm is 74.2%, which also exceeds the other algorithms in terms of F1 measure.

### 4.4. Text Recognition Result

Given that the RCTW-17 dataset covers Traditional and Simplified Chinese characters, uppercase and lowercase English letters, we simulated this characteristic in our generated dataset. In this section, Average Edit Distance (AED) and Normalized Edit Distance (NED) [34] are used to evaluate the recognition results. A low AED indicates better performance. NED is treated as the official ranking metric and is formulated as follows:(7)$$Νοrm=1-\frac{1}{N}\overset{N}{\underset{1}{\sum }}D\left(s_{i},{\hat{s}}_{i}\right)∕max\left(s_{i},{\hat{s}}_{i}\right)$$
where *D*(:) stands for the Levenshtein Distance between $s_{i}$ and ${\hat{s}}_{i}$, which denote the predicted text line in a string and the corresponding ground truths in the regions, respectively. As the norm approaches 1, the performance increases. Table 3 presents the comparison between our method and the other methods. The results show the competitiveness of the proposed recognition module. Figure 9 shows the recognition results of our method in (a) a single text region, (b) a multi-scale text region, (c) a plural text region, and (d) a multi-angle text region. Figure 10 shows several failed recognition cases. Figure 10a shows that the detection is correct, while the text recognition itself is incorrect. In Figure 10b, the incorrect text region location due to the incompleteness of its fan-shape results in the incorrect recognition. In fact, occlusion and missing are a major difficulty in scene text recognition. Figure 10c shows a recognition error caused by recognizing a deeper texture as a text region.

### 4.5. Inverse Experiment

Figure 10c shows that the background of the image affects the text detection. Thus, we invert the generated dataset by rendering the background of the image to black and texts to white. Figure 11 shows the comparison between the original generated dataset and its inverse. Based on previous studies [9], dataset ICDAR2019 Art, LSVT, and ReCTs serve as the training set. In addition, the generated dataset and its inverted version are added in the training phase respetively, and then the verification is performed on RCTW-17. Table 4 lists the results. “Ours (generated)” indicates the training on datasets including ICDAR2019 and the generated dataset, while “ours (inversed)” means the training on dataset including ICDAR2019 and the inverted dataset. The results show that inverted images with less interference, such as background texture, effectively improve the recognition accuracy.

## 5. Robot Object Recognition Experiments

After improving the model recognition accuracy by generating datasets and their inverse, we apply the trained model to the Nao robot to enable its recognition of different types of household objects with similar shapes, similar objects with different shapes, and different types of objects with different shapes. Nao is a humanoid robot developed by Aldebaran Robotics, as shown in Figure 12. Nao has two 2D cameras to capture images and can recognize some basic shapes, objects, and even people. Nao has also four directional microphones and speakers to interact with humans, and it performs speech recognition and dialogue available in 20 languages, including English, French, and Chinese. Therefore, Nao is especially applicable as the research platform of the home service robot. 

The robot extracts keywords from the user’s instruction and then captures images of the objects. The texts on objects are detected and recognized by the model, and then the texts of the recognized objects are stored in a text file. According to the user’s instruction, the confidence of current object is achieved by a fuzzy search in the text file. The object with the highest confidence is the target. Fuzzy search is done by a simple string matching algorithm between keywords and the content of text file. It makes it possible that an object can be represented by different keywords in a user’s instruction; only if the keyword can be searched from the text file, the object can be recognized. For example, “Tango Waffles”, the user can give the instruction either “Tango” or “Waffles”. Figure 13 shows part samples of the test dataset. Pictures are taken by the robot from daily life such as soap, medicine, biscuit, juice, skin care, and so on. The words on the items include Chinese and English. 

Table 5 shows a recognition result for “WAFFLE”. After analysis of the text file, several incorrect recognition words are found and marked by red fonts. The main reason is that one object may have too many texts, most of which are very small. Fortunately, these incorrect texts are not key information for identifying objects. Therefore, experiments on different confidence thresholds of text boxes are carried out. The confidence threshold is higher and recognition results are more accurate, that is, with less incorrect words. However, this approach may miss useful information. Figure 14 shows the relation of the confidence threshold, accuracy, and losing key information from 45 test samples. 

Table 6 shows the recognition results of two samples according to different confidence thresholds. When the confidence threshold is set to 0.97, a trade-off occurs between accuracy and information loss, where red fonts represent the incorrect words. When the confidence threshold is set to 0.98, sample 1 loses “酒精” and sample 2 loses “Virgin” key information.

Table 7 provides several recognition results with a 0.97 confidence threshold. Words marked by gray are keywords that identify objects. The recognition results show that objects can be recogized by these keywords. Table 8 lists some typical examples with the same or similar shapes. For the first two rows, they have a similar shape, but they must be recognized accuately; otherwise, it is dangerous for humans. For the last two rows, two pair images have exactly the same shape. There is no other choice to distinguish them except for text reading. Therefore, the proposed method can recognize arbitrary-shaped and arbitrary-class objects accurately.

Figure 15 demonstrates a robot recognizing the objects. The robot asks, “What do you want?” and we reply, “农夫山泉”. The robot recognizes the characters one by one until it finds the specific object and then points to the object. The water, beer, and drink (upper three objects) have similar shapes, but the robot can accurately distinguish them. 

The experiment results show that the robot equipped with the proposed method can effectively realize accurate object recognition in home environments.

## 6. Conclusions

This paper proposes a human-like object recognition method based on scene text reading technology. The robot recognizes daily objects by reading its texts. The recognition technique of most of the current object recognition methods is based on shapes, which causes issues with objects that have the same or similar shapes. Furthermore, these objects may need accurate recognition, such as medicine or food. Therefore, recognition methods based on shapes is insufficient. We adopt a text detection and recognition model [9], which is trained using ICDIR 2019, comprising a generated comprehensive text dataset and its inverse. After the model is trained by these datasets, the recognition accuracy is improved by 1.26%. Given that objects may have small and less useful texts, a confidence threshold is used as a filter. Through an analysis of confidence experimental results, we set the confidence threshold as 0.97. Subsequently, the recognition results include key information and reduce incorrect recognition due to fine texts. The proposed method can accurately recognize arbitrary-shaped objects. The user can send different commands such as “金典牛奶”, “金典”, or “牛奶”, and the robot can match the recognized text through fuzzy queries. In future work, we can further modify the recognition model by adding NLP technology and regenerate the dataset to train more languages and more complex scene texts. Thus, the practicability of the model can be improved.

## Figures and Tables

**Figure 1 sensors-21-01919-f001:**
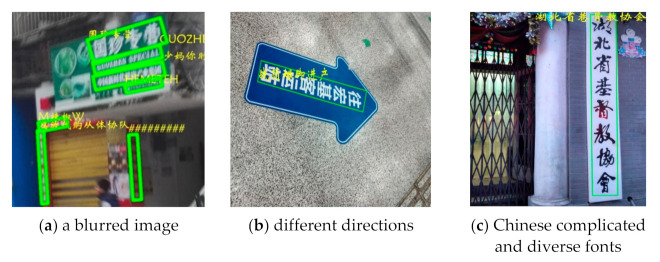
Failed recognition cases (Yellow fonts are the recognition results).

**Figure 2 sensors-21-01919-f002:**
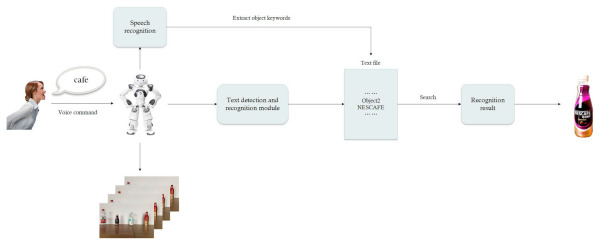
Object recognition framework of robot.

**Figure 3 sensors-21-01919-f003:**
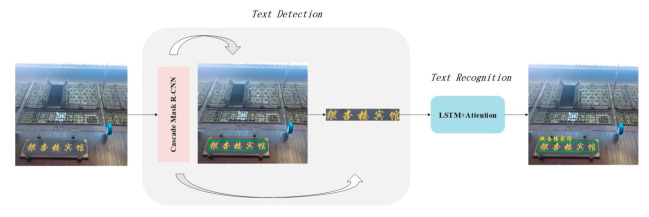
Framework of text detection and recognition.

**Figure 4 sensors-21-01919-f004:**
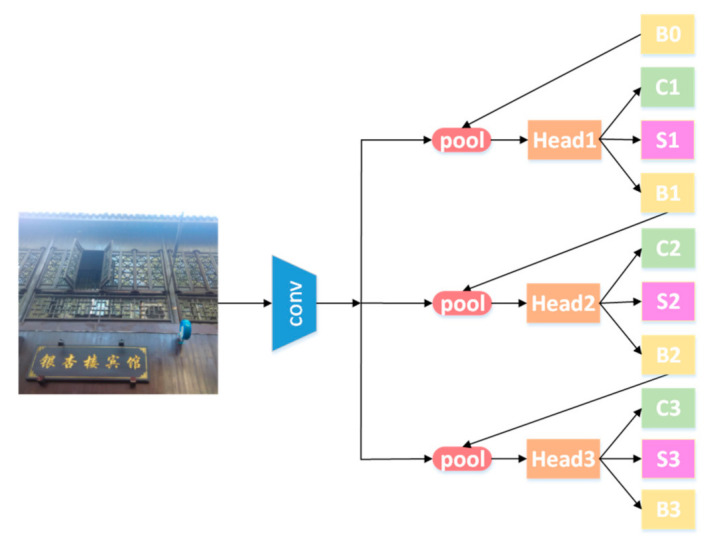
Text detection model based on Cascade Mask Region Convolution Neural Network (R-CNN).

**Figure 5 sensors-21-01919-f005:**
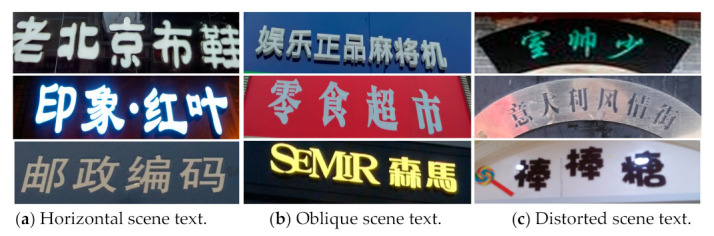
Comparison of different scene texts.

**Figure 6 sensors-21-01919-f006:**
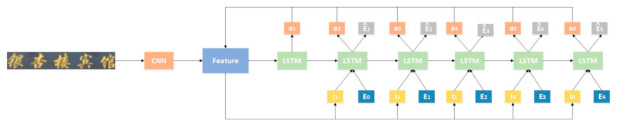
Text recognition model based on Long Short-term Memory (LSTM) and Attention Mechanism.

**Figure 7 sensors-21-01919-f007:**
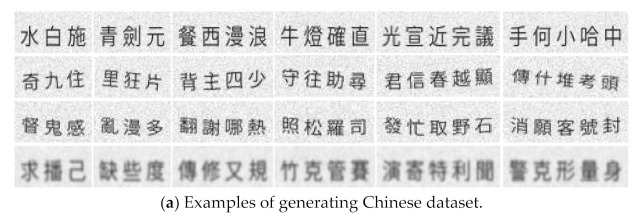
Examples of generating dataset, including Chinese, uppercase, and lowercase letters.

**Figure 8 sensors-21-01919-f008:**
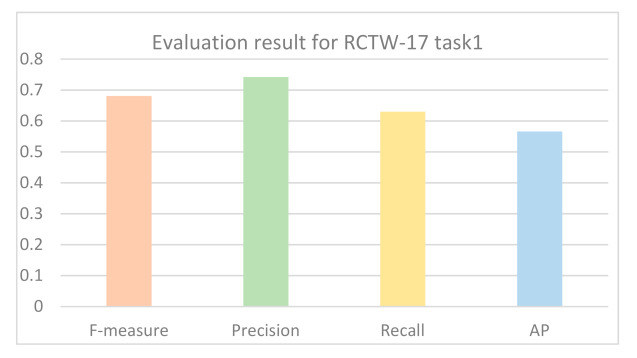
Histogram of experimental results from the RCTW-17 competition platform.

**Figure 9 sensors-21-01919-f009:**
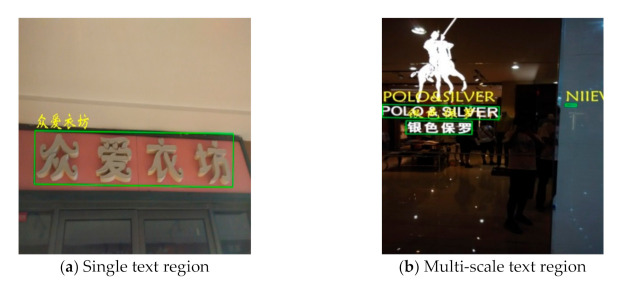
Correct recognition results on RCTW-17.

**Figure 10 sensors-21-01919-f010:**
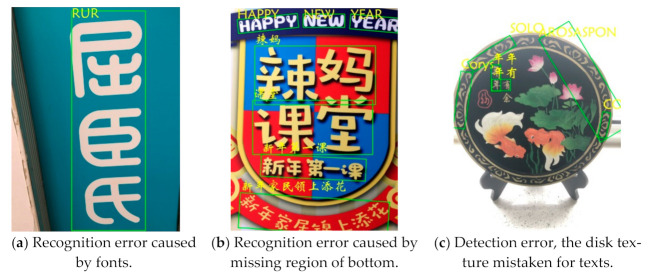
Failed recognition cases.

**Figure 11 sensors-21-01919-f011:**
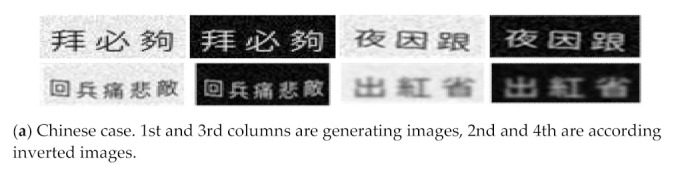
Contrast samples of generated and inverse datasets.

**Figure 12 sensors-21-01919-f012:**
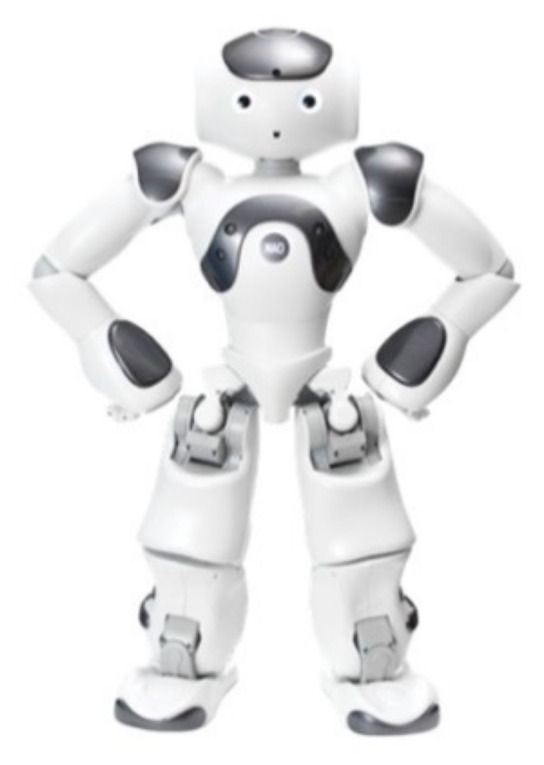
Nao robot.

**Figure 13 sensors-21-01919-f013:**
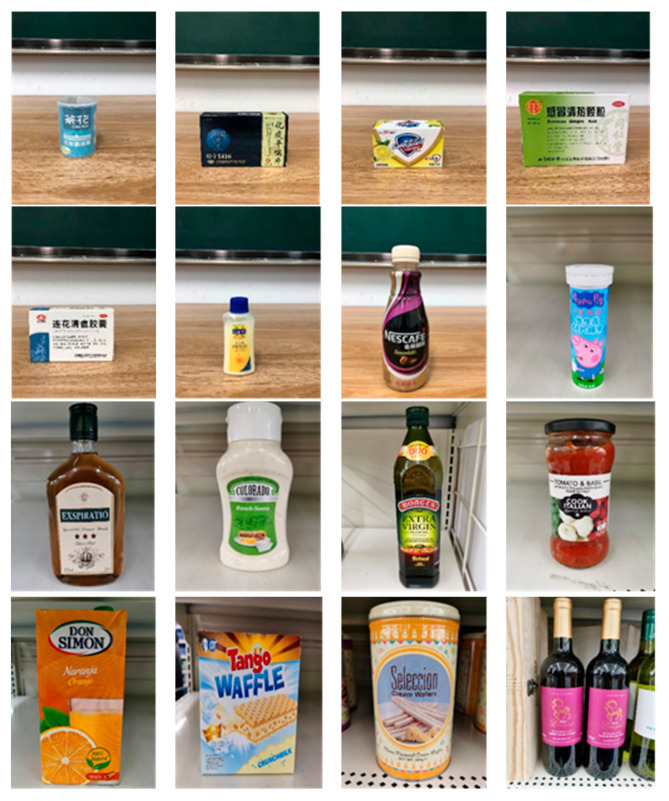
Part of test samples.

**Figure 14 sensors-21-01919-f014:**
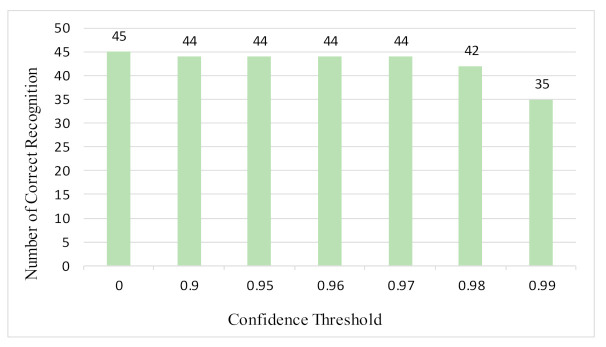
Confidence threshold comparison. There are 45 test samples; with the increase of confidence threshold, the number of correct recognition objects is reducing.

**Figure 15 sensors-21-01919-f015:**
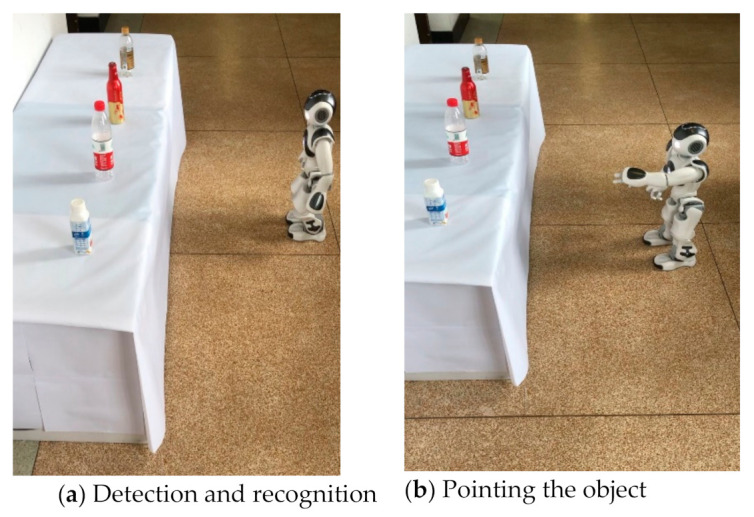
Nao is recognizing objects.

**Table 1 sensors-21-01919-t001:** The number of different kinds of texts from the generating dataset.

Word Count	1	2	3	4	5	6	7	8	9	10	Total
	**Chinese**										
standard	1000	1000	1000	1000	1000	1000	1000	1000	1000	1000	10,000
oblique	5000	5000	5000	5000	5000	5000	5000	5000	5000	5000	50,000
distorted	1000	1000	1000	1000	1000	1000	1000	1000	1000	1000	10,000
obscure	100	100	100	100	100	100	100	100	100	100	1000
	**English (uppercase)**										
standard	1000	1000	1000	1000	1000						5000
oblique	2000	2000	2000	2000	2000						10,000
distorted	1000	1000	1000	1000	1000						5000
obscure	200	200	200	200	200						1000
	**English (lowercase)**										
oblique	2000	2000	2000	2000	2000						10,000

**Table 2 sensors-21-01919-t002:** Comparison results on ICDAR2017 RCTW.

Methods	Precision %	Recall %	F1-Measure%
EAST [35]	59.7	47.8	53.1
RRD [36]	72.4	45.3	55.1
LOMO [37]	80.0	50.8	62.3
TextMountain [38]	80.8	55.2	65.6
IncepText [39]	78.5	56.9	66.0
Border (DenseNet) [40]	78.2	58.5	67.1
End2End-PSL-MS [41]	81.7	57.8	67.7
Ours	74.2	63.0	68.1

**Table 3 sensors-21-01919-t003:** End-to-end recognition results comparison on RCTW-17datasets.

Methods	Average_dist	Normalized %
End2End [41]	27.5	72.9
Attention ocr [9]	26.3	74.2
End2End-PSL [41]	26.2	73.5
**Ours**	25.6	74.9

**Table 4 sensors-21-01919-t004:** Results comparison on inverse dataset.

Methods	Normalized %
Baseline	76.65
Ours (generating)	76.68
Ours (inverse)	77.91

**Table 5 sensors-21-01919-t005:** Recognition result of sample (Red fonts are incorrect recognition texts).

Image	Recognition Result
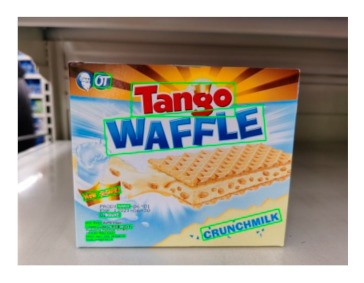	WAFLE/Tango/CRUNCHMILK/New/REGIPE/OT/012704/Jakarta/INDONESIA/PRIMA/Content/Wafer/ABADI/Barat/Milk/Product/2020/Contentl Contenu: 20x89/ULTRA/Halanta Barat 11850-INDONESIA

**Table 6 sensors-21-01919-t006:** Recognition texts on different threshold comparison (Red fonts are incorrect recognition texts).

Image	Confidence ≥ 0	Confidence ≥ 0.9	Confidence ≥ 0.97	Confidence ≥ 0.98
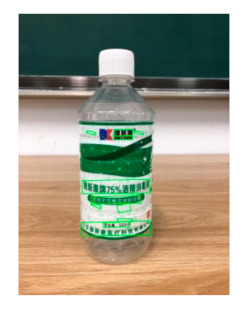	德新康 DEXCON 德新康牌7596酒精消毒瓶 适用于完整皮肤的消费 净含量:500mL sonsloe 山东德新康医疗科技有限公 Sonsloe x.c 1bsbbM ygulondbsloia	德新康 DEXCON 德新康牌7596酒精消毒瓶 适用于完整皮肤的消费 净含量:500mL sonsloe 山东德新康医疗科技有限公	德新康 DEXCON 德新康牌7596酒精消毒瓶	德新康 DEXCON
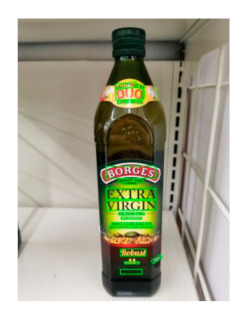	BORGES DUO Robust EXCLUSIVE MEDITERRANEAN EXTRA CAP DRESSING DRESSING COOKING VIRGIN Cold OLIVE SINCE OIL 750n Extraction MASTERS BOROES INTENSE 1896 PATENEDINOUE INTENSITY INTENSE DRESSING	BORGES DUO Robust EXCLUSIVE MEDITERRANEAN EXTRA CAP DRESSING DRESSING COOKING VIRGIN Cold OLIVE SINCE OIL 750n Extraction MASTERS BOROES	BORGES DUO Robust EXCLUSIVE MEDITERRANEAN EXTRA CAP DRESSING DRESSING COOKING VIRGIN	BORGES DUO Robust EXCLUSIVE MEDITERRANEAN EXTRA CAP

**Table 7 sensors-21-01919-t007:** Recognition results with 0.97 confidence threshold (Words marked in gray are keywords that identify objects).

Text detection result	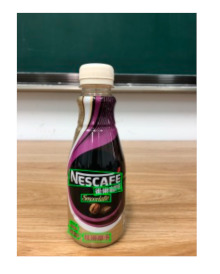	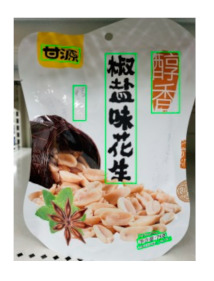	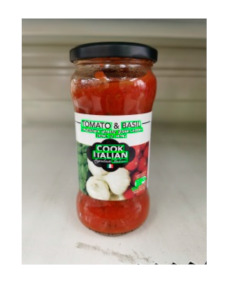	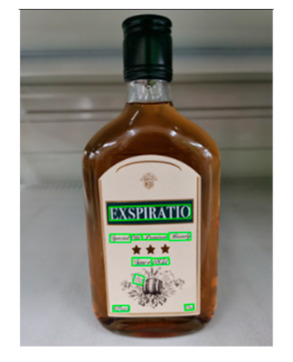
Text recognition result	丝滑摩卡 Smoollatté 雀巢咖啡 NESCAFE	甘源 椒盐味花生 坚果与籽类食品	TOMATO SAUCE PASTA ITALIAN	EXSPIRATIO Biandy

**Table 8 sensors-21-01919-t008:** Recognition results with similar shapes (Words marked in gray are keywords that identify objects).

Image 1	Image 2	Description
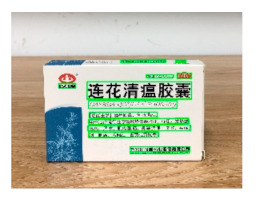	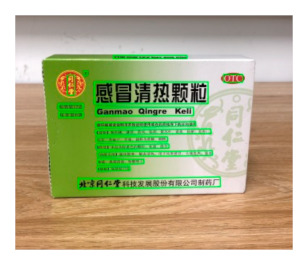	Different Medicine.
连花清盒胶襄	感冒清热颗粒	
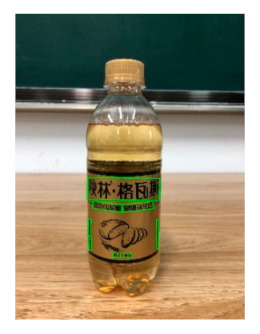	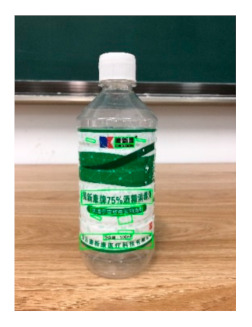	Different Liquid: Left is a drink and right is alcohol.
秋林·格瓦斯	德新康牌7596酒精消毒瓶	
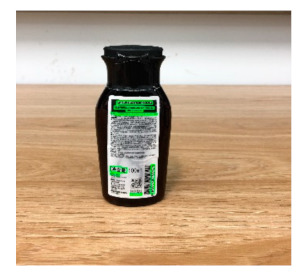	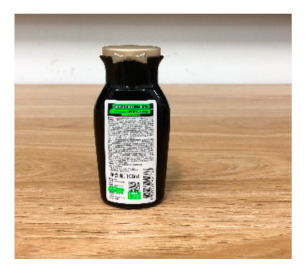	Hair care: left is conditioner and right is shampoo.
吕臻参换活御时生机护发乳	吕臻参换活御时生机洗发水	
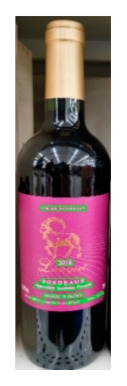	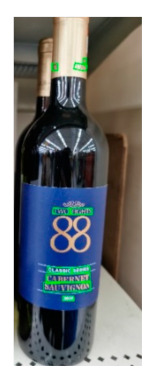	Different brand Wine: left is BORDEAUX and right is CABERNET.
BORDEAUX	CABERNET	

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
