# Peer review of "A Robot Object Recognition Method Based on Scene Text Reading in Home Environments"

_sensors, 2021, doi:10.3390/s21051919_

Round 1

Reviewer 1 Report

In the reviewed paper, the authors proposed an object recognition technique in a robot object application in a home environment. In general, the paper proposes an interesting solution, but some issues should be improved:

1) Used bibliography should be updated. Introduction and related work sections need to analyze the current state of research. The related works need a complex analysis of image recognition solutions. Some discussion about CNNs, rCNNs, masked rCNNs, learning transfer, hybrid solutions like adaptive genetic algorithm for image analysis with a cascade of convolution architectures, the application of heuristic in image analysis also must be discussed. This topic is a basic idea that is used in the proposal.
2) Improve the quality of the images. The used font is too small.
3) Add a mathematical model of processing images in your proposal and training algorithm.
4) Add more comparisons with the latest solutions from the last 2-3 years.
5) Add comparison with transfer learning models.
6) Fig. 8 shows a small histogram but it was prepared in other software. Please recreate all figures/charts in one software for better reception by the readers.
7) In the conclusions add more discussion about the novelty and the impact on the other, current state of research.

Reviewer 2 Report

The work describes an object recognition method based on a proposed technique for text detection and recognition. The work is interesting, and the results seem to be relevant in relation to the state-of-the-art. In general, the manuscript explained in a completed way the applied techniques. Results are consistent and the experiments were well supported statistically. The work represents an interesting approach to improve the current lack of recognition of human-like recognition objects. However, some concerns have to be addressed in order to improve the understanding of the manuscript.

The introduction is a bit confused since the second paragraph included an uncommon order of ideas. The objective of the work is stated in the second paragraph and unconnected with the main contributions at the end of the first section. I suggest reordering ideas for an easy reading and a better understanding of the purpose of the work. For example, across the document, a question about why big text from databases was used to validate the technique was only answered in the 5th section, when the technique was validated with objects that can be grabbed by a robot. A general idea can be clearer from the methodology.

Also, the Methodology seems to be fragmented among the following sections. Some methods and technique descriptions are explained in the 4th (Experiments on detection and recognition models), and 5th (Robot object recognition Experiments) sections. It makes it difficult to understand the real scope of the work. 

Some other minor concerns:

- Line 36. Even if the robot has been developed by authors themselves, it can be cited in previous works. Moreover, a wide description of the current and previous works would clarify the objective of the work.

- For easy reading, I suggest defining some abbreviations in the text (e.g. LSTM, RPN) or including categorical words to describe them (e.g. Word2vec, InceptionV4). 
- Line 76. As a description of the contribution of the work, please indicate the modes from reference 9.
- Figure 4. “I” is missing in the Figure. 
- Equation 1. Please, include the representation of Ltotal. 

- Line 212. Some text datasets used in the work are mentioned. However, as Chinese is not widely known all over the world, I suggest introducing briefly as "databases".
- Line 245. "task1" is ambiguous. Please define. Moreover, AP obtained (0.5664) is related to Figure 8, but the corresponding bar shows a result higher than 0.6. It seems to be inconsistent. In the same Figure, the correct term is"AP" or "Average Precision" instead of "Precision"?

Some figures include a very small text for reading.

Reviewer 3 Report

Minor improvements in the text are recommended like following

1. Abstract line 14: What is 0.4? Is it a factor or percentage?

2. Abstract at line 19~20, the sentence should be phrased as "Then, the object with the maximum confidence is considered to be/selected as the target"

3. Introduction at line 36, the sentence should be "which enables the robot to perceive the environment more efficiently"

4. Introduction at line 38, the term should be "cultured person" or "civilized person"

5. Introduction at line 38, Figure 1 shows the failed cases instead of the framework for robot recognition. The figure with the framework in Figure 2.

6. Introduction line 63: It is unclear that why these images are presented as failed cases, while these pictures show that text regions are detected successfully. Also, the recognition result in Yellow color is not readable. I would be better if the result is also written below each image. So the reader may be able to see if it failed at correct recognition of text.

7. Introduction line 82: The sentence should be phrased as "Higher confidence threshold results in more accurate recognition"

8. In the abstract, it is mentioned that not only the overall accuracy is improved but the F-measure of text detection is also improved by 0.4. However, in the introduction and related work, all the previous work is simply listed like a catalog. Was there a problem faced in previous methods of text-detection which was overcome to achieve this better performance? If yes please present the related work with their pros and cons. If this improvement is attributed purely to the dataset you generated then please re-write the abstract in such a way that this statement is clearly understood.

9. In Methodology at line 142~143: The first sentence used the name cascade R-CNN and the second one uses the name mask R-CNN. It creates confusion whether they are separate modules or one module named Cascade Masked R-CNN as shown in Figure 3. If you are referring to the different branches of one bigger module then modify the diagram to show the part played by each branch or re-write the statements in a more clear way.

10. In Text Detection, Figure 4: Head 1, Head 2, and Head 3 all produce the same outputs (B3,S,C3). Should they produce (B1,S1,C1), (B2,S2,C2) and (B3,S3,C3) respectively?

11. In the Text Detection, the order of equations (1), (2), and (3) should be reversed as each equation uses the output of the latter. Similarly, the equation of smoothL1 should be explained first, then the equation of LCa should be explained and the equation of Ltotal should be explained at the end. In the original text the explanation of Equations (2) and (3) is mixed. Some part of equation (2) is explained before equation (3) and some part is explained after equation (3). The order of presenting the equations and their respective explanation is confusing.

12. In Arbitrary Shaped Scene Text Recognition, at line 190: It is unclear that predicted word embedding uses the correct version of the symbol (capitalized E)? Please review it and correct it if it is by mistake.

13. In Dataset at line 212: References to the datasets need to be provided as below.

    1. C. Chng, Y. Liu, Y. Sun, et al, “ICDAR 2019 Robust Reading Challenge on Arbitrary-Shaped Text-RRC-ArT”, in Proc. of ICDAR 2019.
    2. Yipeng Sun, Zihan Ni, et al. “ICDAR 2019 Competition on Large-scale Street View Text with Partial Labeling – RRC-LSVT” In Proc. Of ICDAR 2019.
    3. Shangbang Long, Jiaqiang Ruan, et al. “TextSnake: A Flexible Representation for Detecting Text of Arbitrary Shapes” in ECCV 2018
    4. Baoguang Shi, Cong Yao, et al. “ICDAR2017 Competition on Reading Chinese Text in the Wild (RCTW-17)” In Proc of ICDAR 2017

14. In Text Detection Results, Figure 8 seems quite unnecessary the whole figure represents only 3 numbers. Moreover, a histogram is made between values of the same nature, while Precision, Recall, and F-measure are different types of results that should be compared to Precision, Recall, and F-measures of other works or papers not with each other.

15. Inverse experiment at lines 281 and 282. It is unclear which dataset was used for training? Line 281 says that the ICDAR2019 dataset was used at the training set. While line 282 says that your generated dataset and inverted versions are used for training.

16. Robot object recognition Experiment, at line 309: Rephrase the sentence as “It makes it possible a that an object can be”

17: Table 7, Line 341~349: Line numbers are inserted in the Table.

18: After reading the paper, it is unclear that 1.26% improvement as mentioned in the abstract at line 15 and conclusion is in comparison the object detection with ICDAR2019 dataset based training or it is presented as compared to normal object detection models without using text recognition.

19 It is suggested that a table should be provided to compare the object recognition performance as compared to the methods that don’t use text recognition.

20. It will be helpful if the captions of figures are a little more descriptive. One should be able to understand the figure based on its caption. Instead in this paper, you have to refer to the main text which is sometimes on a different page and often leads to confusion.

In the light of the review above, I don’t find any major problem with the paper. All the improvements are suggested to improve the quality of the paper.

Round 2

Reviewer 1 Report

It can be accepted in the current form.

Reviewer 2 Report

I thank the authors by accept the suggestions from the review.